 **eLIFE**

# Active torque generation by the actomyosin cell cortex drives left–right symmetry breaking

**Sundar Ram Naganathan**[1,2,3†], **Sebastian Fürthauer**[2,3†‡], **Masatoshi Nishikawa**[1,2,3], **Frank Jülicher**[2], **Stephan W Grill**[1,2,3]*

[1]Biotechnology Center, Technical University Dresden, Dresden, Germany; [2]Max Planck Institute for the Physics of Complex Systems, Dresden, Germany; [3]Max Planck Institute of Molecular Cell Biology and Genetics, Dresden, Germany

**Abstract** Many developmental processes break left–right (LR) symmetry with a consistent handedness. LR asymmetry emerges early in development, and in many species the primary determinant of this asymmetry has been linked to the cytoskeleton. However, the nature of the underlying chirally asymmetric cytoskeletal processes has remained elusive. In this study, we combine thin-film active chiral fluid theory with experimental analysis of the *C. elegans* embryo to show that the actomyosin cortex generates active chiral torques to facilitate chiral symmetry breaking. Active torques drive chiral counter-rotating cortical flow in the zygote, depend on myosin activity, and can be altered through mild changes in Rho signaling. Notably, they also execute the chiral skew event at the 4-cell stage to establish the *C. elegans* LR body axis. Taken together, our results uncover a novel, large-scale physical activity of the actomyosin cytoskeleton that provides a fundamental mechanism for chiral morphogenesis in development.

*For correspondence: stephan.grill@biotec.tu-dresden.de

†These authors contributed equally to this work

Present address: ‡Courant Institute of Mathematical Sciences, New York University, New York, United States

## Introduction

Most organisms are bilaterally asymmetric with morphologically distinct left and right hand sides. Bilateral asymmetry of organisms, organs, and tissues emerges early in development and is dependent on chiral symmetry breaking of cells and subcellular structures (*Hayashi and Murakami, 2001*; *Shibazaki et al., 2004*; *Danilchik et al., 2006*; *Xu et al., 2007*; *Hejnol, 2010*; *Tamada et al., 2010*; *Vandenberg and Levin, 2010*; *Savin et al., 2011*; *Taniguchi et al., 2011*; *Wan et al., 2011*; *Huang et al., 2012*). In many species the primary determinant of chirality has been linked to the cytoskeleton with both the microtubule (*Nonaka et al., 1998*; *Ishida et al., 2007*) and the actomyosin cytoskeleton (*Danilchik et al., 2006*; *Hozumi et al., 2006*; *Spéder et al., 2006*) (AD Bershadsky, personal communication, November 2013) playing prominent roles. Generally, how chiral molecules and chiral molecular interactions generate chiral morphologies on larger scales remains to be a fundamental problem (*Turing, 1952*; *Brown and Wolpert, 1990*; *Henley, 2012*). For example, it has been observed that myosin motors can rotate actin filaments in motility assays (*Sase et al., 1997*; *Beausang et al., 2008*). Yet, it remains unknown which types of large-scale mechanical activities arise from such types of chiral molecular interactions. In this study, we describe that the actomyosin cytoskeleton can generate active torques at cellular scales, and that the cell uses active torques to break chiral symmetry.

## Results and discussion

We investigated chiral behaviours of the actomyosin cell cortex in the context of polarizing cortical flow in the 1-cell *Caenorhabditis elegans* embryo (*Munro et al., 2004*; *Mayer et al., 2010*). The cell cortex, sandwiched between the membrane and cytoplasm, is a thin actin gel containing myosin

**eLife digest** Most living things have left and right sides that are not identical. A well-known example of this 'left–right asymmetry' is the position of the human heart within the human body. While the human heart is always on the left, in other situations it is possible for either the left side or the right side to be preferred: for example, some people prefer to write with their right hand, while others prefer to write with their left hand.

In animals, left–right asymmetry starts early in the development of the embryo. A structure in cells called the cytoskeleton is known to be responsible for generating the asymmetry in many species. The cytoskeleton is mostly made of two types of proteins—rod-like proteins called microtubules and filaments of a protein called actin—but it is not clear how it is involved in establishing left–right asymmetry.

The cytoskeleton has many functions in the cell: for example, it maintains the shape of the cell, it splits the contents of the cell during cell division, and it transports various things around inside the cell. The cytoskeleton is constantly moving and changing shape: all this activity involves another protein called myosin that binds to the actin filaments and moves along them to generate pulling forces.

Naganathan et al. studied newly fertilized embryos of the nematode worm *Caenorhabditis elegans* when they contained just one cell. The experiments showed that myosin can generate turning forces that twist the actin cortical layer, leading to local rotations in the cytoskeleton that make the cell asymmetrical. This is controlled by a group of proteins called Rho proteins.

Next, Naganathan et al. studied embryos that contained four cells. Again, myosin generates local rotations in the cytoskeleton, which are involved in setting up left–right body direction in this stage of development. These experiments show that changes in the cytoskeleton of individual cells can drive asymmetry in the whole embryo. The next challenge will be to understand how myosin is controlled so that rotations only occur during specific cell divisions.

motors and actin binding proteins (**Pollard and Cooper, 1986**; **Clark et al., 2013**). Given the chirality of cortical constituents, we first asked if cortical flow breaks chiral symmetry. We quantified the cortical flow velocity field $v$ using particle image velocimetry in *C. elegans* zygotes containing GFP-tagged non-muscle myosin II (NMY-2) (**Mayer et al., 2010**). Flow proceeds primarily along the anteroposterior (AP) axis ($x$-direction), however, we also observed flow vectors to have a small component in the direction orthogonal to the AP axis ($y$-direction). Notably, the posterior and anterior halves of the cortex counter-rotate relative to each other (**Figure 1A,B**, **Figure 1—figure supplement 1**, **Video 1**), with $y$-velocities of ~–2.5 µm/min and ~1 µm/min respectively (**Figure 1D**). We define the chiral counter-rotation velocity $v_c$ as the difference between spatially averaged $y$-velocities in the posterior and the anterior region (**Figure 1B**) and measured $v_c$ at 858 time points during flow in 25 embryos. We find that the distribution of $v_c$ is shifted towards negative values, with a mean of −2.9 ± 0.3 µm/min (mean ± error of mean at 99% confidence unless stated otherwise, **Figure 1C**). Thus, counter-rotating cortical flow breaks chiral symmetry at the 1-cell stage, with the posterior half undergoing a counterclockwise rotation when viewed from the posterior pole (**Figure 1A**). Notably, chiral counter-rotating flow precedes the previously reported chiral whole-cell rotation of the zygote during cell division (**Schonegg et al., 2014**).

Since AP flow depends on myosin activity (**Mayer et al., 2010**), we asked if chiral flow does so as well. We tested if reducing myosin activity through RNAi of the myosin regulatory light-chain *mlc-4* reduces the chiral counter-rotation velocity $v_c$. We found that 8 hrs of *mlc-4 (RNAi)* not only reduces the AP flow velocity (**Munro et al., 2004**) but also significantly reduces $v_c$ (Wilcoxon rank sum test at 99% confidence; mean: −1.1 ± 0.4 µm/min, **Figure 1C**, **Video 2**) when compared to non-RNAi embryos. We conclude that chiral flow depends on myosin activity.

We next sought to understand how myosin activity can drive both AP and chiral flow. We pursue the idea that molecular-scale torque generation (**Sase et al., 1997**; **Beausang et al., 2008**) leads to the emergence of active torques on larger scales and make use of a physical description of the cell cortex as a thin film of an active chiral fluid (**Fürthauer et al., 2012, 2013**). In our description, force and torque generation at the molecular scale give rise to both an active contractile tension $T$ and an active torque

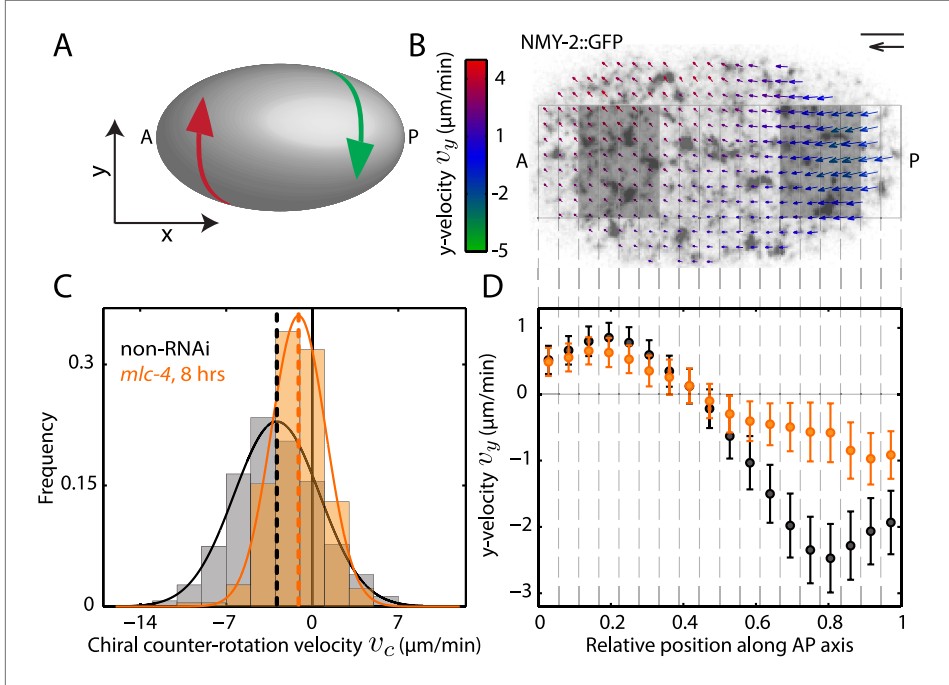

**Figure 1**. Chiral flow depends on myosin activity. (**A**) Sketch of a *C. elegans* embryo. Curved arrows illustrate chiral counter-rotating flow in the anterior (A, red) and posterior (P, green) half of the embryo, respectively. (**B**) Time-averaged cortical flow field (arrows) at the bottom surface of a representative *C. elegans* embryo viewed from the outside of the embryo in this and all other images. Arrow colors indicate *y*-velocity. Scale bar, 5 μm. Velocity scale arrow, 20 μm/min. (**C**) Histogram of instantaneous chiral counter-rotation velocity $v_c = \langle v_y \rangle_P - \langle v_y \rangle_A$, where $\langle v_y \rangle_A$ $\left( \langle v_y \rangle_P \right)$ is the average of the *y*-component of the velocity *v* over the left (right) shaded area in (**B**), for non-RNAi (858 frames from 25 embryos; gray) and *mlc-4 (RNAi)* (8 hrs; 223 frames from 7 embryos; beige). Dashed vertical lines indicate mean $v_c$. (**D**) *y*-velocity $v_y$ along the AP axis averaged over 18 vertical stripes as indicated, for non-RNAi (black, averaged over 25 embryos) and *mlc-4 (RNAi)* (beige, averaged over 7 embryos). Error bars, SEM.

The following figure supplement is available for figure 1:

**Figure supplement 1**. Similar flow fields were obtained when imaging actin as compared to imaging myosin
*Video 8*.

---

density *τ* (**Figure 2A**) to drive cortical flow. Under conditions of azimuthal symmetry (**Figure 2—figure supplement 1**, appendix), the AP flow velocity ($v_x$) and the *y*-velocity ($v_y$) obey the equations of motion,

$$\partial_x T = \eta \partial_x^2 v_x - \gamma v_x$$

$$\partial_x \tau = \frac{1}{2} \eta \partial_x^2 v_y - \gamma v_y,$$  (1)

where *η* is the 2D viscosity of the cortical layer and *γ* quantifies friction with membrane and cytoplasm (**Mayer et al., 2010**). From the structure of **Equation 1**, we see that gradients in active tension *T* along the AP axis drive AP flow, while gradients in active torque density *τ* along the AP axis drive chiral flow orthogonal to the AP axis (**Figure 2B**, bottom sketch). We introduce the chirality index *c* = *τ*/*T*, which quantifies their relative magnitude. We assume that both active tension *T* and active torque density *τ* are proportional to the local myosin concentration, leading to a single value of the chirality index *c* that is constant over the embryo. This remains a useful approximation even for cases where *T* and *τ* exhibit more complex dependencies on myosin concentration or where they are independently regulated (see below). In such cases the single value of *c* we determine corresponds to an average, capturing the overall chirality index of the embryo (see appendix). Accordingly, we calculated the theoretical AP and

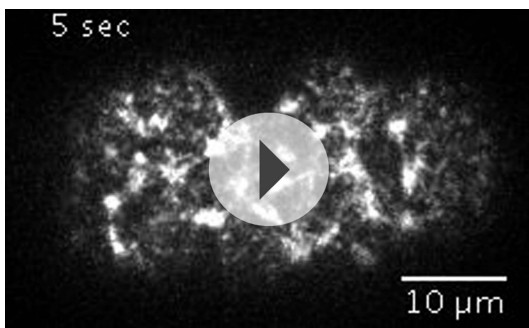

**Video 1**. Cortical flow breaks chiral symmetry. Cortical flow during AP polarization of the *C. elegans* zygote exhibits chiral behaviors with the posterior and the anterior halves of the cortex counter-rotating relative to each other.

chiral flow profiles from the experimentally determined myosin distribution and found a best match with the experimental profiles for a hydrodynamic length of $\ell = \sqrt{\eta/\gamma} = 16 \pm 0.6$ μm (*Mayer et al., 2010*) and an overall chirality index of $c = 0.58 \pm 0.09$ (*Figure 2B*; see appendix). We conclude that a significant part of myosin activity is utilized for generating active torques. The handedness of active torques is clockwise when viewed from the outside of the embryo as indicated by the positive sign of the chirality index $c$. When considering the observed AP myosin gradients, active torques of this handedness give rise to counterclockwise flow in the posterior domain when viewed from the posterior tip, see *Figure 2B* for an illustration.

We next sought to investigate if changing myosin activity affects the overall chirality index. To this end, we performed a series of mild-to-stronger (*Baggs et al., 2009*) *mlc-4 (RNAi)* experiments with feeding times of 4, 6, and 8 hrs, respectively, and determined $c$ for each condition. We refer to this as weak perturbation RNAi experiments as we aim to identify principle phenotypical alterations upon a mild deviation from non-RNAi conditions, similar to determining the linear response to a small perturbation. While AP flow velocity $v_x$ and the chiral flow velocity $v_c$ were generally reduced at 4 and 6 hrs of *mlc-4 (RNAi)* (*Figure 3A*, *Figure 3—figure supplement 1–3*, *Video 3*), $c$ remained unchanged from non-RNAi conditions ($c$, 0.61 ± 0.07 at 4 hrs and 0.52 ± 0.06 at 6 hrs of RNAi, compared to 0.58 ± 0.09 for non-RNAi; *Figure 3A*, *Figure 3—figure supplement 3*). However, 8 hrs of *mlc-4 (RNAi)* not only resulted in a large reduction of both AP and chiral flow velocities but also led to a significant reduction of $c$ (0.14 ± 0.04, *Figure 3A*, *Figure 3—figure supplement 3*). This indicates that the overall ratio of active torque density to active tension is not changed by weak reduction of *mlc-4* activity but is altered at stronger RNAi conditions when cortical structure is affected (*Figure 3—figure supplement 4A* and *Video 2*).

We next asked whether there are conditions that modify active torque generation without affecting active tension. To this end, we tested if small changes in Rho signaling, which regulates myosin activity as well as actin dynamics (*Maekawa et al., 1999*), have a different impact on AP and chiral flow and thus change $c$. We performed a series of mild-to-stronger RNAi of the Rho GEF *ect-2* and the Rho GAP *rga-3*. We found that weak perturbation RNAi of *ect-2* led to a substantial decrease in chiral but not AP flow and thus a decrease in the overall chirality index $c$ when compared to non-RNAi conditions (*Figure 3A,D*; see also *Figure 3—figure supplement 4B*, *Video 4*). Conversely, weak perturbation RNAi of *rga-3* led to a substantial increase in chiral but not AP flow and thus an increase in the overall chirality index $c$ (*Figure 3A,D*; see also *Figure 3—figure supplement 4C*, *Video 5*). Thus, a weak perturbation of *ect-2* and *rga-3* affects chiral but not AP flow, unlike a weak perturbation of *mlc-4* which affects both. We conclude that a principle phenotypical alteration upon mild modifications of Rho pathway activity is a change of the chirality index, or, in other words, mild modifications of Rho pathway activity change active torque generation without affecting active tension.

We next asked whether actomyosin active torques participate in bilateral symmetry breaking, since this requires a chiral process. In *C. elegans*, embryonic handedness is determined at the 4-cell stage when the ABa and ABp cells skew clockwise by ~20° (as viewed dorsally in the AP–LR plane, *Figure 4A*) (*Wood, 1991*; *Bergmann et al., 2003*). We first tested if the clockwise skew in ABa is accompanied by chiral cortical flow. Strikingly, we observed chiral cortical flow in ABa, with the

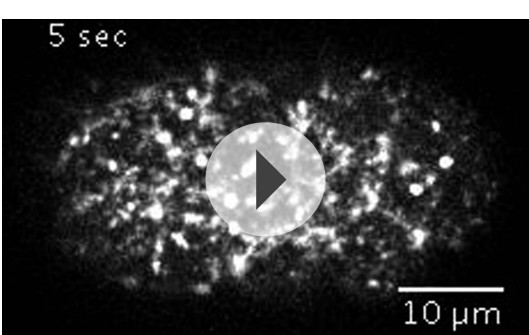

**Video 2**. Chiral flow depends on myosin activity. 8 hrs of *mlc-4 (RNAi)* leads to a substantial reduction of both AP and chiral flow.

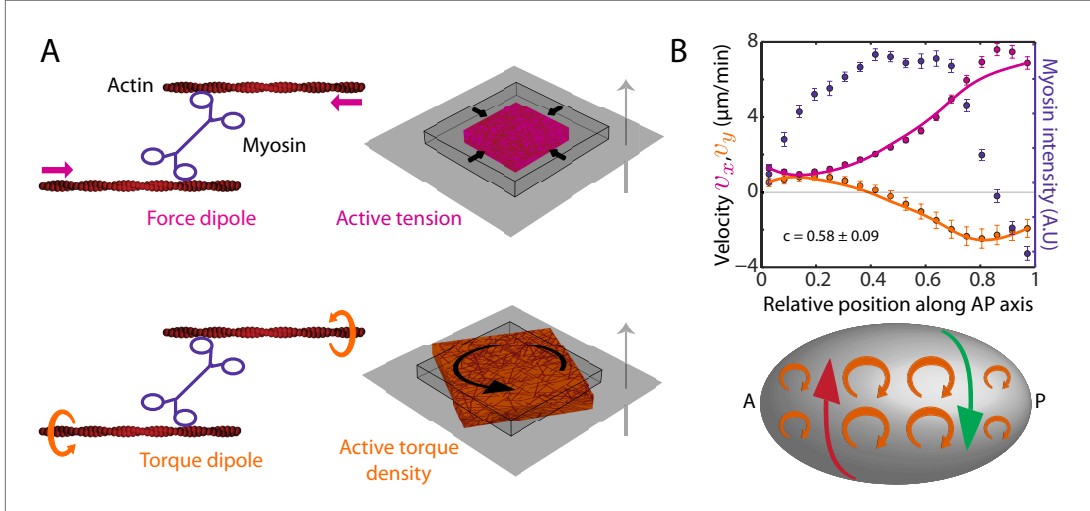

**Figure 2**. The cortex actively generates torques. (**A**) Left, myosin heads consume ATP to pull (**Kron and Spudich, 1986**) and twist (**Sase et al., 1997**; **Beausang et al., 2008**) actin filaments, leading to the generation of a force dipole (top, magenta) and a torque dipole (bottom, beige). Right, these can generate an active tension and an active torque density at larger scales, causing an isolated piece of cortex to contract (top) and rotate (bottom). Gray surface, membrane; cube with wire frames, non-contracted (non-rotated) piece of cortex; magenta (beige) cubes, contracted (rotated) piece of cortex. The gray arrow points from the outside to the inside of the cell and the rotation is clockwise when viewed from the outside. (**B**) Top, myosin intensity (blue markers) and velocity profiles (magenta markers, AP flow velocity $v_x$; beige markers, $y$-velocity $v_y$) along the AP axis (**Figure 1B,D**) for the non-RNAi condition (averaged over 25 embryos). Error bars, SEM. Magenta and beige curves, respective theoretical velocity profiles ($c = 0.58 \pm 0.09$). Bottom, sketch of a *C. elegans* embryo with clockwise active torques in beige (as viewed from the outside of the embryo). A gradient in myosin concentration along the AP axis (see plot above) leads to a gradient in active torques (shown here with varying sizes of the clockwise torques), resulting in a chiral flow (red and green arrows) orthogonal to the gradient.

The following figure supplement is available for figure 2:

**Figure supplement 1**. Myosin distribution is azimuthally symmetric.

cortex in both future daughter cells counter-rotating ($v_c = -5.2 \pm 1.1$ μm/min, mean ± error of mean at 95% confidence, **Video 6**). The handedness of chiral flow is identical to that at the 1-cell stage, indicative of a presence of active torques with the same sign of $c$. If these chiral counter-rotating flows participate in the clockwise skew of both daughter cells, we would expect that changing active torque generation should affect the chiral skew at the 4-cell stage. To this end, we performed weak perturbation RNAi of the Rho pathway members, *ect-2* and *rga-3*, to specifically modify active torques. We first tested whether chiral flows are affected at the 4-cell stage under these conditions. We found that 4.5 hrs of *ect-2 (RNAi)* led to a significant decrease in chiral flow velocity, $v_c$ ($-3.4 \pm 1.4$ μm/min), while 4.5 hrs of *rga-3 (RNAi)* led to a significant increase in $v_c$ in the ABa cell ($-6.7 \pm 0.7$ μm/min, **Figure 4B**, **Video 6**), similar to our observations at the 1-cell stage. We next tested whether changing chiral flow velocity at the 4-cell stage is concomitant with a change in the degree of clockwise skew. Indeed, we found that 4.5 hrs of *ect-2 (RNAi)* led to a significant decrease in skew (15.8° ± 4.9°) in the ABa cell measured in the AP–LR plane, while 4.5 hrs of *rga-3 (RNAi)* led to a significant increase in skew (37.8° ± 6.1°) when compared to non-RNAi conditions (23.6° ± 3.7°; **Figure 4A** and **Figure 4—figure supplement 1**). Similar results were obtained in ABp (**Figure 4—figure supplement 1**). Thus, changing counter-rotating chiral flow velocity in these cells by weak perturbation of the Rho pathway leads to a change in the degree of skew. This suggests that active torque generation and chiral counter-rotating flow participate in the execution of the LR symmetry breaking chiral skew event at the 4-cell stage.

Finally, we tested whether genes that affect establishment of the L/R body axis impacts chiral flow. To investigate this, we quantified chiral flow velocities and the overall chirality index $c$ at the 1-cell stage under conditions of RNAi of the Wnt signaling genes *dsh-2*, *gsk-3*, *mig-5*, *mom-2*, and *mom-5*,

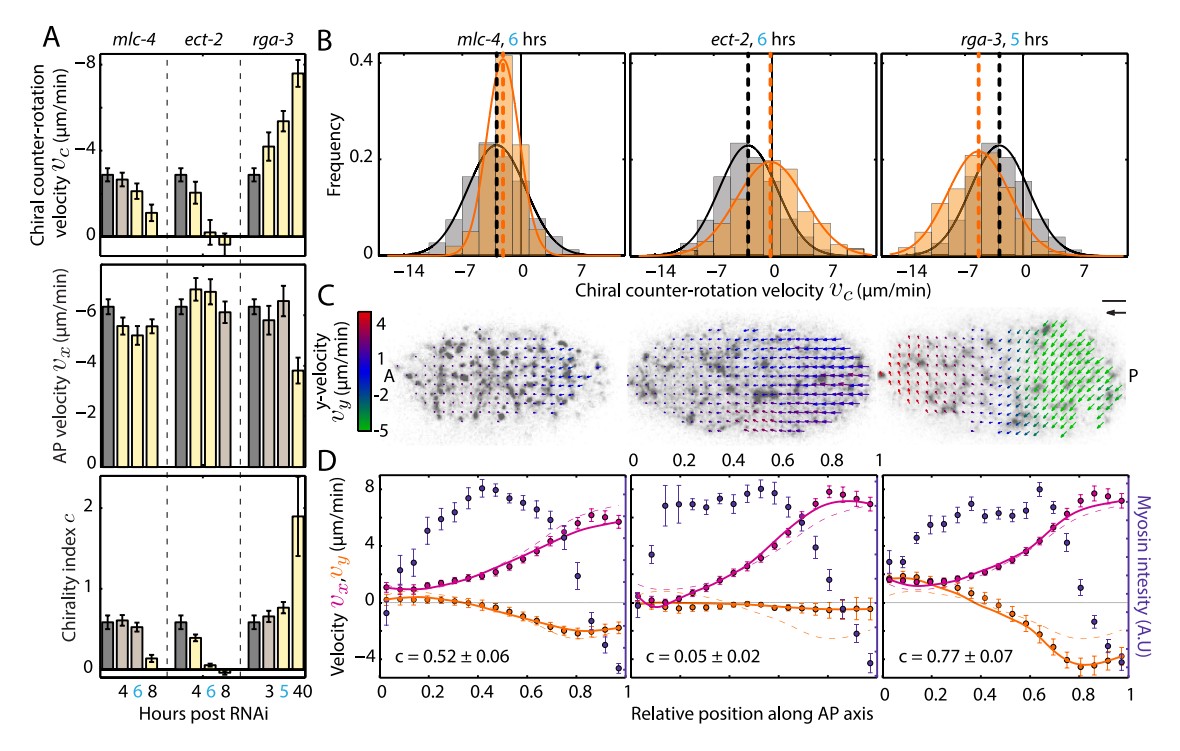

**Figure 3**. Ratio of active torque to active tension is modulated by Rho. (**A**) Chiral counter-rotation velocity $v_c$ (top), AP velocity $v_x$ (middle), and chirality index $c$ (bottom) for non-RNAi (gray), *mlc-4* (4, 6, 8 hrs RNAi), *ect-2* (4, 6, 8 hrs RNAi), and *rga-3* (3, 5, 40 hrs RNAi). Error bars, error of the mean with 99% confidence. Yellow bars, significant difference to non-RNAi condition; brown bars, no significant difference. (**B**) Histogram of instantaneous chiral counter-rotation velocity $v_c$ for *mlc-4* (left; 6 hrs; 235 frames from 7 embryos), *ect-2* (middle; 6 hrs; 338 frames from 9 embryos), and *rga-3* (right; 5 hrs; 402 frames from 10 embryos) RNAi. Gray histograms, non-RNAi condition. Dashed lines, mean $v_c$ (**C**) Respective time-averaged cortical flow field (arrows) of representative embryos (gray, myosin). Arrow colors indicate $y$-velocity $v_y$. Scale bar, 5 μm. Velocity scale arrow, 20 μm/min. (**D**) Respective average myosin intensity (blue markers) and velocity profiles (magenta markers, AP flow velocity $v_x$; beige markers, $y$-velocity $v_y$) along the AP axis for each RNAi condition. Error bars, SEM. Magenta and beige curves, respective theoretical velocity profiles. Dashed lines, non-RNAi theoretical velocity profiles.

The following figure supplements are available for figure 3:

**Figure supplement 1**. Chiral counter-rotation velocity $v_c$ for *mlc-4*, *ect-2*, and *rga-3* RNAi.

**Figure supplement 2**. AP velocity $v_x$ for *mlc-4*, *ect-2*, and *rga-3* RNAi.

**Figure supplement 3**. Theoretical velocity profiles for *mlc-4*, *ect-2* ,and *rga-3* RNAi.

**Figure supplement 4**. Comparison of cortical structure through imaging GFP-tagged NMY-2.

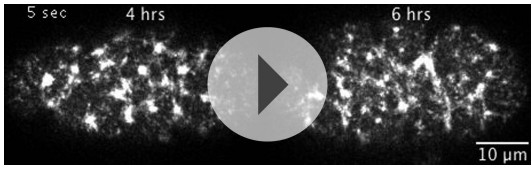

**Video 3**. Chirality of the cortex is unaffected under weak perturbation of myosin activity. 4 and 6 hrs of *mlc-4 (RNAi)* leads to a proportional change of AP and chiral flow, with the chirality index remaining unchanged under these conditions.

which are known to regulate aspects of bilateral symmetry breaking (*Walston et al., 2004*; *Pohl and Bao, 2010*). Strikingly, we found that all these conditions (except *mom-5*) led to reduced chiral flow and a significant reduction of the overall chirality index $c$ at the 1-cell stage (*Figure 4C*, *Figure 4—figure supplement 2–4*, *Video 7*). These results are indicative of a fundamental link between genes that affect LR symmetry breaking and chiral counter-rotating flow. Since Wnt-induced signals in many systems propagate through Rho GTPases to promote morphological changes

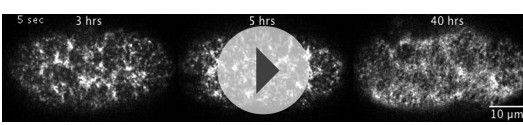

**Video 4**. Chiral flow decreases with decreasing Rho activity. *ect-2 (RNAi)* leads to a substantial reduction in chiral flow with a minimal change in AP flow.

**Video 5**. Chiral flow increases with increasing Rho activity. *rga-3 (RNAi)* leads to a substantial increase in chiral flow with a minimal change in AP flow.

(*Schlessinger et al., 2009*), we speculate that these effects are propagated through Rho signaling (*Figure 3*). Taken together, our results indicate that active torque generation and chiral counter-rotating flows participate in the establishment of the L/R body axis of *C. elegans*.

To conclude, the actomyosin cytoskeleton in *C. elegans* generates active chiral torques with clockwise handedness when viewed from the outside of the cell. They drive a specific pattern of chiral flows which can be understood quantitatively based on the physics of active gels with chiral asymmetries (*Kruse et al., 2005*; *Fürthauer et al., 2012, 2013*). Furthermore, our weak perturbation RNAi experiments indicate that Rho activity affects cortical chirality in a way that does not depend on its role in activating myosin. On the one hand, this raises an interesting question whether active chiral torques arise directly from chiral interactions between actin and myosin (*Figure 2A*) or whether they rather emerge through myosin molecular force generation and non-trivial tension–torque coupling (*Gore et al., 2006*; *De La Cruz et al., 2010*) in the actomyosin network. On the other hand, through these weak perturbation RNAi experiments we have identified specific conditions under which cortical chirality and active torques can be selectively modified. Bilateral symmetry breaking requires a chiral process, and we used these specific conditions to demonstrate that in *C. elegans*, this chiral process could be provided by active chiral torque generation of the actomyosin cortical layer for driving the spindle skew at the 4-cell stage. We note that a plausible scenario for driving spindle skew by counter-rotating flows is similar to the rotation that a crawler excavator or a digger can execute on the spot. This is done by such a machine rotating its two chains in opposite directions. In our context, the chain rotations correspond to the counter-rotating flows and the rotation of the machine corresponds to the spindle rotation giving rise to the skew.

Our results imply that active torques are generated at multiple stages during development, in the zygote during polarity establishment, without immediate consequences with respect to LR symmetry breaking, and again at the 4-cell stage, but here as an instructional and mechanistic event that helps to break left/right symmetry. Chiral morphogenetic rearrangements have been observed at other stages in *C. elegans* development (*Pohl and Bao, 2010*) and during the first cleavage (*Schonegg et al., 2014*; *Singh and Pohl, 2014*), as well as in other systems (*Shibazaki et al., 2004*; *Danilchik et al., 2006*; *Géminard et al., 2014*). It is interesting to speculate that all these events might be driven by active torque generation in the actomyosin layer. As such, our work paves the way for a mechanistic understanding of chiral morphogenesis of cells, tissues, and organisms.

## Materials and methods

### *C. elegans* strains

The following transgenic lines were used in this study: TH455 *(unc-119(ed3) III; zuIs45[nmy-2::NMY-2::GFP + unc-119(+)] V; ddIs249[TH0566(pie1::Lifeact::mCherry:pie1)])* for imaging cortical flow, LP133 *(nmy-2(cp8[NMY-2::GFP + unc-119(+)]) I; unc-119(ed3) III)* (*Dickinson et al., 2013*) for imaging counter-rotation of AB cells, and SWG003 *(nmy-2(cp8[NMY-2::GFP + unc-119(+)]) I; unc-119(ed3) III; gesIs002[unc-119(ed3) III; (pie-1::Lifeact::tagRFP-T::pie-1 + unc-119(+))])* for quantifying chiral flow fields with an actin probe. For imaging the chiral skew event at the 4-cell stage, a mCherry::Histone; mCherry::PH-PLC1δ1 transgenic line was generated by crossing OD70 (*Kachur et al., 2008*) to a line expressing Moesin::GFP and mCherry::Histone obtained from the Piano lab (New York University, New York, USA). *C. elegans* worms were cultured on OP50-seeeded NGM agar plates as described (*Brenner, 1974*).

### RNA interference

RNAi experiments were performed by feeding (*Timmons et al., 2001*). Worms were placed on feeding plates (NGM agar containing 1 mM isopropyl-*β*-D-thiogalactoside and 50 µg ml⁻¹ ampicillin) and

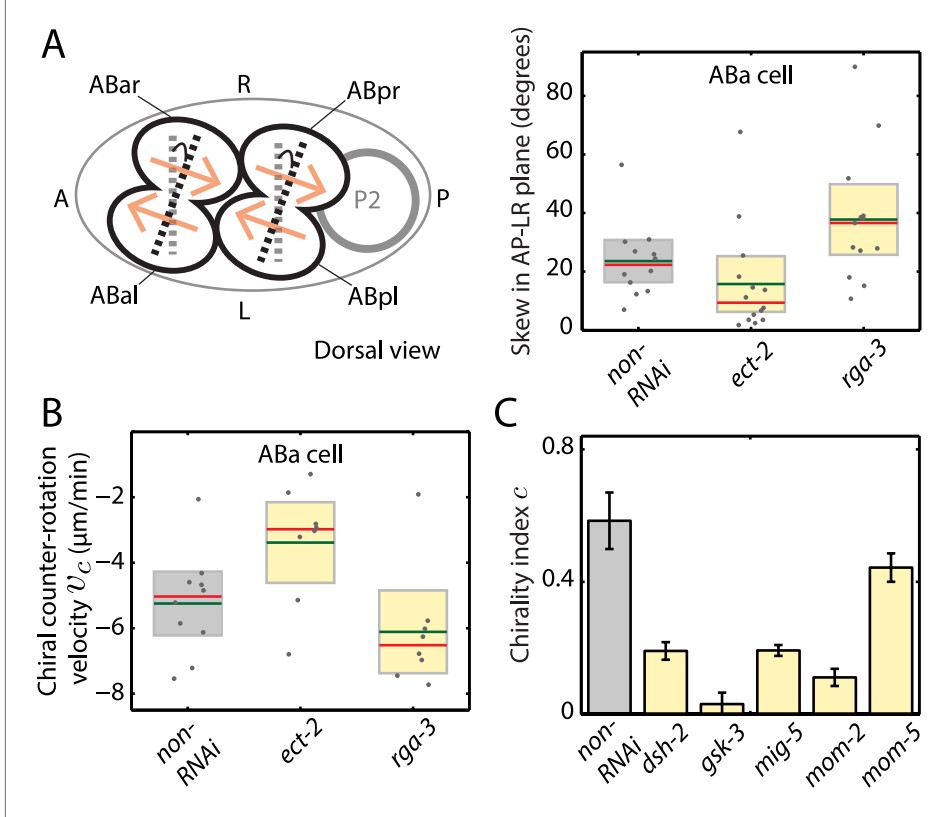

**Figure 4**. Active torques participate in *L/R* body axis establishment. (**A**) A schematic of the skew angle measurement in the AP–LR plane. Gray dashed line, initial nuclei position; black dashed line, skewed nuclei position; beige arrows, direction of cortical flow on the dorsal surface (***Video 6***). To the right are the chiral skew angles of ABa for non-RNAi (gray), *ect-2 (RNAi)* (4.5 hrs) and *rga-3 (RNAi)* (4.5 hrs) in the AP–LR plane. Gray circles, skew angle in individual videos; shaded areas, SEM; green horizontal lines, mean skew angle; red horizontal lines, median skew angle; yellow shaded areas, knockdown conditions with a significant difference (95% confidence with the Wilcoxon rank sum test) from the non-RNAi condition. (**B**) Chiral counter-rotation velocity $v_c$ for non-RNAi (gray), *ect-2 (RNAi)* (4.5 hrs) and *rga-3 (RNAi)* (4.5 hrs) quantified at the 4-cell stage during ABa cytokinesis. Note that one outlier was removed for computing mean $v_c$ for *rga-3 (RNAi)*. The expected flow profiles from our theoretical description, given a stripe of high myosin activity (corresponding to the cleavage plane), is shown in ***Figure 4—figure supplement 5***. (**C**) Overall chirality index *c*, for non-RNAi (gray) and for Wnt signaling genes (40 hrs RNAi) that impact the establishment of the *L/R* body axis. Interestingly, *gsk-3* not only results in a reduced chiral counter-rotation velocity but also in an increased AP velocity (***Figure 4—figure supplements 2–4***). Error bars, error of the mean with 99% confidence. Yellow bars, significant difference to non-RNAi condition; brown bars, no significant difference.

The following figure supplements are available for figure 4:

**Figure supplement 1**. Chiral skew quantifications during bilateral symmetry breaking of the organism.

**Figure supplement 2**. Chiral counter-rotation velocity $v_c$ for RNAi of Wnt signaling genes.

**Figure supplement 3**. AP velocity $v_x$ for RNAi of Wnt signaling genes.

**Figure supplement 4**. Theoretical velocity profiles for RNAi of Wnt signaling genes.

**Figure supplement 5**. Theoretical velocity profiles for a stripe of high myosin activity.

---

incubated for the specified number of hours at 25°C. We defined feeding time (number of hours of RNAi) as the time between transfer of worms to the feeding plate and putative fertilization of the egg. Worms were dissected in M9 buffer and the embryos were mounted on 2% agarose pads for image

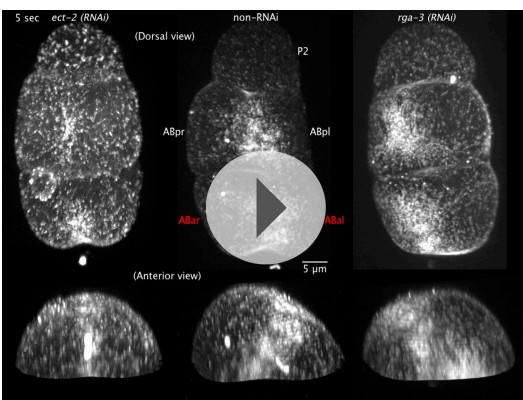

**Video 6**. Chiral flow accompanies the LR symmetry breaking skew event at the 4-cell stage. Dorsal view of a representative 4-cell stage embryo, with ABa and ABp cells exhibiting counter-rotating cortical flow during cytokinesis, for 4.5 hrs of *ect-2 (RNAi)*, non-RNAi and 4.5 hrs of *rga-3 (RNAi)*. Anterior view of these Videos is shown at the bottom for visualizing counter-rotation in ABa cell. Flashing cyan arrows indicate the direction of counter-rotating cortical flow. Note that counter-rotation of AB cells is significantly reduced in *ect-2 (RNAi)* and significantly increased in *rga-3 (RNAi)* compared to the non-RNAi condition. Quantification of chiral flow velocities (**Figure 4B**) was performed in the ABa cell (marked in red).

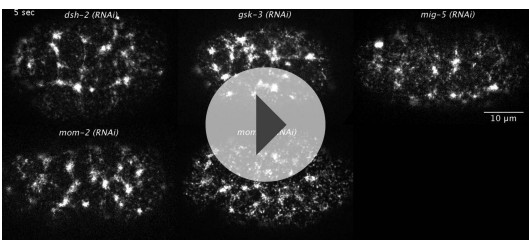

**Video 7**. Wnt signaling genes regulate chiral flow. RNAi of Wnt signaling genes leads to a substantial reduction in chiral flow.

acquisition. *rga-3* feeding clone was obtained from Ahringer lab (Gurdon institute, Cambridge, United Kingdom), *ect-2* and *mlc-4* from Hyman lab (MPI-CBG, Dresden, Germany). Feeding clones *dsh-2*, *gsk-3*, *mig-5*, *mom-2*, and *mom-5* were obtained from Source Bioscience (Nottingham, United Kingdom).

For performing weak perturbation RNAi experiments (from 3–12 hrs of RNAi), L4 staged worms were first incubated overnight on OP50 plates at 25°C. Young adults were then transferred to respective RNAi feeding plates. For performing 40 hr RNAi experiments, early L4 staged worms were directly transferred to respective RNAi feeding plates and incubated at 25°C.

## Image acquisition

All videos were acquired at 23–24°C, with a spinning disc confocal microscope using a Zeiss C-Apochromat 63X/1.2 NA objective lens and a Yokogawa CSU-X1 scan head. The following emission filter was used for all acquisitions unless specifically stated: 525/50 nm bandpass filter from Semrock (Rochester, New York). Micromanager software (Vale lab, UCSF) was used to acquire videos using the Hamamatsu ORCA-flash camera.

Confocal videos of cortical NMY-2::GFP for non-RNAi, *mlc-4*, *ect-2*, and *rga-3 (RNAi)* were acquired using an Andor iXon EMCCD camera (512 by 512 pixels). A stack consisting of three z-planes (0.5 µm apart) with a 488 nm laser and an exposure of 150 ms was acquired at an interval of 5 s from the onset of cortical flow until the first cell division. The maximum intensity projection of the stack at each time point was then subjected for further analysis.

Confocal videos of cortical NMY-2::GFP for *dsh-2*, *gsk-3*, *mig-5*, *mom-2*, and *mom-5 (RNAi)* were acquired using an Andor Neo sCMOS camera (2560 by 2160 pixels). A stack consisting of two z-planes (0.5 µm apart) with a 488 nm laser and an exposure of 150 ms was acquired at an interval of 5 s from the onset of cortical flow until the first cell division. The maximum intensity projection of the stack at each time point was then subjected for further analysis.

Chiral skew at the 4-cell stage was imaged by using mCherry::Histone; mCherry::PH-PLC1δ1 dual transgenic line. Confocal videos were acquired using a Hamamatsu ORCA-flash 4.0 camera (2048 by 2048 pixels). A stack consisting of 25–30 z-planes (1 µm apart) with a 561 nm laser and an exposure of 300 ms (emission filter – 641/75 nm bandpass filter from Semrock) was acquired at an interval of 30 s from metaphase of the AB lineage at the 4-cell stage until telophase of the AB lineage at the 8-cell stage.

Counter-rotation of the AB cells was imaged using the LP133 strain. Embryos at the 4-cell stage were first identified and an eye-lash tool was then used to rotate the embryo to obtain a dorsal view. Confocal videos from the dorsal side of the embryo were then acquired using a Hamamatsu ORCA-flash 4.0 camera (2048 by 2048 pixels). A stack consisting of 25 z-planes (0.5 µm apart) with a 488 nm laser and an exposure of 100 ms was acquired at an interval of 5 s from the start of telophase of the AB lineage at the 4-cell stage until cytokinesis.

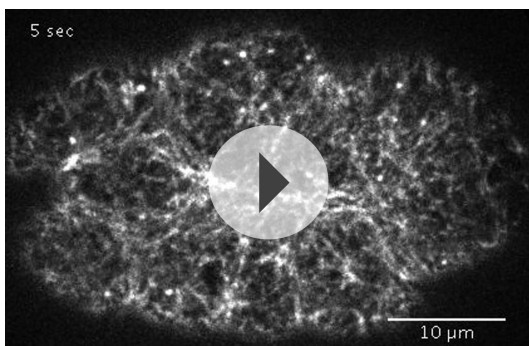

**Video 8**. Chiral flow observed with an actin probe. Cortical flow visualized through Lifeact::tagRFP-T exhibits similar chiral behaviors.

## Flow velocity analysis

2D cortical flow velocity fields were obtained by performing Particle Image Velocimetry (PIV) (*Raffel et al., 2007*) using the freely available PIVlab MATLAB algorithm (pivlab.blogspot.de). PIVlab was employed by performing a 3-step multi pass (with linear window deformation), where the final interrogation area was 16 pixels with a step of 8 pixels.

To obtain the flow profiles, 2D velocity fields were projected to the AP axis by dividing the embryo into 18 bins along the AP axis (*Figure 1B*), and by spatially averaging the x-component or the y-component of velocity along each bin in a single frame. The average velocity in each bin was then averaged over time across the entire flow period (from start of the flow till pseudocleavage). These time-averaged flow profiles were then averaged across all embryos for one experimental condition. Bin extent in the y direction was restricted to a stripe of about 13 µm (*Figure 1B*). An accurate quantification of chiral flow fields is only possible when the flow axis is approximately aligned with the long axis of the embryo, and we removed from our analysis a small number of embryos (3/28 embryos for the non-RNAi condition) which in the bottom plane analysis appeared to clearly polarize from the side.

Chiral counter-rotation velocity $v_c$ was quantified in each frame by subtracting the y-component of velocity in the anterior (spatially averaged across bins 3 to 6, *Figure 1B*) from the y-component of velocity in the posterior (spatially averaged across bins 13 to 16, *Figure 1B*). $v_c$ from each frame was computed across the entire flow period from all embryos of one experimental condition and a histogram was plotted.

For quantification of chiral counter-rotation velocity $v_c$ in the ABa cell, the cleavage plane as viewed from the dorsal side of the embryo was first manually identified using FIJI. PIV was then performed and the component of velocity vectors parallel to the cleavage plane was calculated. $v_c$ was then computed by subtracting velocity components in the right daughter cell (spatially averaged across a box of width 5 µm close to the cleavage plane) from the velocity components in the left daughter cell (spatially averaged across a box of size 5 µm close to the cleavage plane). $v_c$ from each frame was computed across the counter-rotation flow period and a time-averaged $v_c$ was reported. We removed from our analysis one video each from non-RNAi and *ect-2 (RNAi)* conditions and 2 videos from *rga-3 (RNAi)* condition that resulted in a marked whole body rotation along the DV-LR plane.

## Chiral skew analysis

Chiral skew analysis was performed by analyzing the multi-stack videos using Imaris v7.6.5 and v7.7 software (Bitplane, Zurich, Switzerland). The anterior and posterior pole positions were first manually identified and used to define the AP-vector. The DV-vector was then obtained utilizing the nuclear position of EMS and the AP-vector. The initial and skewed nuclei vectors of ABa and ABp cells at the beginning and end of telophase, respectively, were determined by identifying the corresponding nuclei positions. We used MATLAB to determine the angle between the initial and skewed nuclei vectors of ABa and ABp. We first determined the skew angle by projecting all vectors on to the AP–LR plane (dorsal view, *Figure 4—figure supplement 1A*). This is the plane in which the skew has been reported previously (*Wood, 1991*; *Bergmann et al., 2003*). Next, we determined the skew angle in the DV–LR plane (anterior view, *Figure 4—figure supplement 1B*) by projecting all vectors on to the DV–LR plane. Finally, we also determined the full (3D) angle between the respective initial and skewed vectors.

## Additional information on the chiral skew event at the 4-cell stage

As described in the main text (*Figure 4A*), in the ABa cell, *ect-2 (RNAi)* (4.5 hrs) led to a significantly reduced skew in the AP–LR plane, whereas *rga-3 (RNAi)* (4.5 hrs) led to a significantly increased skew. Similar to the ABa cell, the ABp skew was also significantly reduced under *ect-2 (RNAi)* condition

(*Figure 4—figure supplement 1A*). However, the skew was unchanged in the ABp cell for *rga-3 (RNAi)*. We next determined the unprojected full (3D) skew angle for each condition (*Figure 4—figure supplement 1C*). Intriguingly, we observed that the unprojected full ABp cell skew was marginally increased under *rga-3 (RNAi)* condition (*Figure 4—figure supplement 1C*) even though the projected AP–LR plane skew remained unchanged (*Figure 4—figure supplement 1A*). Similarly, we observed that the unprojected full ABa cell skew was not significantly different for *ect-2 (RNAi)* compared to the non-RNAi condition (*Figure 4—figure supplement 1C*). This difference in skew between the AP–LR projected and the unprojected angles for the same conditions indicated that there could be an additional skew in a different plane. To this end, we determined the skew in the DV–LR plane. Interestingly, we detected an ~20° skew in this plane even for the non-RNAi condition (*Figure 4—figure supplement 1B*). Generally, the difference between the full (3D) skew angles and the AP–LR projected angles is due to this additional skew in the DV–LR plane. This illustrates that the chiral skew at the 4-cell stage is more complex than previously reported, with a chiral rotation in the DV–LR plane in addition to the AP–LR plane.

## Foci size analysis

A characteristic myosin foci size was determined by performing spatial myosin fluorescence intensity autocorrelation in MATLAB. The autocorrelation was performed in a stripe of about 27 μm wide and 13 μm high in the anterior of the embryo and the analysis was carried out in each frame during the first 75 s of cortical flow. The spatial autocorrelation decay determined in each analysis frame was then fitted with a single exponential function and we define the decay length of this fit as the characteristic foci size. The foci size thus determined in each frame was then averaged over all analysis frames for a single embryo and an ensemble average was reported.

## Appendix

### Cortical distribution of myosin is azimuthally symmetric

Through laser ablation experiments, we have shown that cortical flow is driven by gradients in active tension along the AP axis. We consider that active tension is proportional to myosin fluorescence intensity (*Mayer et al., 2010*). Accordingly, we observed that a histogram (*Figure 2—figure supplement 1A*) of the difference in spatially averaged myosin fluorescence intensities between the posterior and anterior halves of the embryo (see shaded areas in *Figure 2—figure supplement 1A*) is shifted towards negative values, indicating an asymmetric myosin distribution along the AP axis. Similarly, it is plausible that chiral flows could originate from a gradient in myosin fluorescence intensity (i.e., myosin activity) along the azimuthal direction. To test for such a gradient, we analyzed the difference in spatially averaged myosin fluorescence intensities between the top and bottom halves of the embryo in the posterior (see shaded areas in *Figure 2—figure supplement 1B*). This analysis was performed in each frame and compiled across the entire flow period from all embryos of the non-RNAi condition. The resulting histogram (*Figure 2—figure supplement 1B*) shows that there is no detectable gradient in myosin fluorescence intensity along the azimuthal direction. Thus, we conclude that the embryo does not appear to break azimuthal symmetry with respect to myosin distribution and that chiral flow does not originate from myosin activity gradients along the azimuthal direction.

### Cortical equation of motion

We derive equations of motion for the cell cortex, describing it as a thin film of active chiral fluid (*Fürthauer et al., 2012, 2013*). We start by stating the conservation of momentum and angular momentum. Momentum conservation in the absence of external forces is expressed by,

$$\partial_t(\rho v_\alpha) = \partial_\beta \sigma_{\alpha\beta}^{tot}, \tag{2}$$

where $\rho v$ is the momentum density, with the mass density $\rho$ and the center of mass velocity $v$, and $\sigma_{\alpha\beta}^{tot}$ is the stress tensor. A summation convention over repeated indices is implied. In general, the stress tensor $\sigma_{\alpha\beta}^{tot} = \sigma_{\alpha\beta} + \sigma_{\alpha\beta}^{a} - P\delta_{\alpha\beta}/3$ can be decomposed into a symmetric traceless part, an antisymmetric part, and a trace part corresponding to the hydrostatic pressure, which we denote as $\sigma_{\alpha\beta}$, $\sigma_{\alpha\beta}^{a}$, and $P$, respectively.

Conservation of angular momentum in the absence of external torques reads,

$$\partial_t\left(I\Omega_{\alpha\beta}\right) = \partial_\gamma M_{\alpha\beta\gamma} - 2\sigma_{\alpha\beta}^{a}. \tag{3}$$

In this study, we introduced the intrinsic or 'spin' angular momentum density $I\Omega_{\alpha\beta}$, with $\Omega_{\alpha\beta}$ and $I$ being the spin rotation rate and the moment of inertia density, respectively. Moreover, we introduced the spin angular momentum flux tensor $M_{\alpha\beta\gamma}$. Note that in general the moment of inertia tensor $I$ is a symmetric fourth rank tensor (**Stark and Lubensky, 2005**; **Fürthauer et al., 2012**). Here, we choose it to be diagonal for simplicity.

We describe the cell cortex as an active chiral fluid with the constitutive equations,

$$\sigma_{\alpha\beta} = \eta u_{\alpha\beta} + \zeta\left(p_\alpha p_\beta - \delta_{\alpha\beta}/3\right), \tag{4}$$

$$\sigma_{\alpha\beta}^{a} = \eta'\left(\Omega_{\alpha\beta} - \omega_{\alpha\beta}\right), \tag{5}$$

$$\frac{1}{2}M_{\alpha\beta\gamma} = \kappa\partial_\gamma\Omega_{\alpha\beta} + \zeta_1\varepsilon_{\alpha\beta\nu}p_\nu p_\gamma, \tag{6}$$

where we introduced the strain rate $u_{\alpha\beta} = (\partial_\alpha v_\beta + \partial_\beta v_\alpha)/2$ and the vorticity $\omega_{\alpha\beta} = (\partial_\alpha v_\beta - \partial_\beta v_\alpha)/2$. The phenomenological coefficients $\eta$, $\eta'$, and $\kappa$ describe the passive response of the material. The coefficients $\zeta$ and $\zeta_1$ quantify active processes. Notably, $\zeta$ quantifies the tendency of molecular motors to generate active shear in the material, whereas $\zeta_1$ quantifies their tendency to produce active angular momentum fluxes. In general, additional couplings to the constitutive equations (**Equations 4–6**) exist, see (**Fürthauer et al., 2012**). In this study, we constrain ourselves to the simplest set of equations which reproduces the phenomenology observed in the *C. elegans* cell cortex. The vector $p$ denotes the broken symmetry of the cortical material in the thin direction and reflects that inhomogeneous protein distributions lead to different boundary conditions with the cytosol and the cell membrane (**Fürthauer et al., 2013**). Thus in the following we consider $p_x = p_y = 0$ and $p_z = 1$, thus $p^2 = 1$.

To obtain equations of motion for the cortical material, we use the constitutive **Equations 4–6** together with the force balance **Equation 2** and the torque balance **Equation 3** at low Reynolds number. The 3D torque and force balances for the cortical material read,

$$\frac{1}{2}(\eta + \eta')\partial_\gamma^2 v_\alpha + \eta'\partial_\beta\Omega_{\alpha\beta} - \partial_\alpha(\zeta - P)/3 + \partial_\beta\zeta p_\alpha p_\beta = 0, \tag{7}$$

$$\kappa\partial_\gamma^2\Omega_{\alpha\beta} + \partial_\gamma\zeta_1\varepsilon_{\alpha\beta\nu}p_\nu p_\gamma = \eta'\left(\Omega_{\alpha\beta} - \omega_{\alpha\beta}\right), \tag{8}$$

where we used the incompressibility condition $\partial_\gamma v_\gamma = 0$. After some algebra we obtain a fourth order differential equation for the velocity field,

$$0 = \frac{1}{2}\eta\partial_\gamma^2 v_\alpha + \partial_\alpha(\zeta - P)/3 + \partial_\beta\left[\zeta p_\alpha p_\beta + \partial_\gamma\left(\zeta_1\varepsilon_{\alpha\beta\nu}p_\nu p_\gamma\right)\right]$$
$$- \frac{\kappa}{2\eta'}\partial_\nu^2\left[(\eta + \eta')\partial_\gamma^2 v_\alpha - \partial_\alpha(\zeta - P)/3 + \partial_\beta\left(\zeta p_\alpha p_\beta\right)\right]. \tag{9}$$

The length scale $\sqrt{\kappa/\eta'}$ is the length on which the intrinsic rotation rate $\Omega_{\alpha\beta}$ decays to the vorticity $\omega_{\alpha\beta}$ and is set by some molecular lengths in the system (**Stark and Lubensky, 2005**; **Fürthauer et al., 2012, 2013**). An upper bound for $\sqrt{\kappa/\eta'}$ is the thickness $d$ of the cell cortex, which is smaller than 1 µm (**Clark et al., 2013**). Thus to describe long ranged flows in the cell cortex, we can consider the limit of $\sqrt{\kappa/\eta'} \to 0$ and obtain,

$$0 = \frac{1}{2}\eta\partial_\gamma^2 v_\alpha - \partial_\alpha(\zeta - P)/3 + \partial_\beta\left[\zeta p_\alpha p_\beta + \partial_\gamma\left(\zeta_1\varepsilon_{\alpha\beta\nu}p_\nu p_\gamma\right)\right]. \tag{10}$$

Using the approximation $\sigma_{zz}^{tot} \approx P^{ext}/3$, valid in thin films where $P^{ext}$ is the constant external pressure, as well as the incompressibility condition $\partial_\gamma v_\gamma = 0$, we can rewrite this expression as,

$$0 = \frac{1}{2}\eta\partial_\gamma^2 v_i + \frac{1}{2}\eta\partial_i\partial_j v_j + \partial_i p_z^2\zeta + \partial_j\partial_z\zeta_1\varepsilon_{ijz}. \tag{11}$$

Here, the roman indices $i$ and $j$ denote the in film directions $x$ and $y$. Finally, we integrate **Equation 11** over the film thickness $d$ and obtain.

$$\frac{1}{2}\eta\partial_j^2\overline{v}_i + \frac{1}{2}\eta\partial_i\partial_j\overline{v}_j - \gamma\overline{v}_i = \partial_i T + \varepsilon_{ij}\partial_j\tau, \tag{12}$$

where we introduced the averaged velocity

$$\overline{v}_i = \frac{1}{d}\int_0^d dz v_i, \tag{13}$$

the active cortical tension

$$T = \frac{1}{d}\int_0^d dz \zeta, \tag{14}$$

the active cortical torque density

$$\tau = \frac{1}{d}\int_0^d dz \partial_z\zeta_1, \tag{15}$$

and the friction coefficient $\gamma$

$$\gamma\overline{v}_i \simeq \frac{1}{d}\eta\partial_z v_i\big|_0^d = \frac{1}{d}\int_0^d dz\eta\partial_z^2 v_i. \tag{16}$$

Since the cell is azimuthally symmetric, the $x$ and $y$ components of the *Equation 12* decouple such that,

$$\ell\partial_x^2\overline{v}_x - \frac{1}{\ell}\overline{v}_x = \partial_x T \quad \text{and} \tag{17}$$

$$\frac{1}{2}\ell\partial_x^2\overline{v}_y - \frac{1}{\ell}\overline{v}_y = \partial_x\tau. \tag{18}$$

Here, we have introduced the hydrodynamic length $\ell = \sqrt{\eta/\gamma}$.

In summary, molecular force generation by actin and myosin is represented by active force and torque dipoles. These give rise to an active tension and an active torque density in the material and can generate flows. Spatial gradients in active tension are generated by a spatially inhomogeneous distribution of myosin motors. These give rise to a flow along the gradient of myosin density (*Mayer et al., 2010*), see *Equation 17*. Spatial gradients in active torque density on the other hand give rise to a flow orthogonal to the gradient of myosin density (*Fürthauer et al., 2013*) (see *Equation 18*) in a direction that is set by the chirality of the torque dipoles in the cell cortex. The ratio of active torque densities to active tension thus generated is quantified by the chirality index,

$$c = \int_0^d dz\partial_z\zeta_1 \bigg/ \int_0^d dz\zeta. \tag{19}$$

A sketch of this mechanism is provided in *Figure 2B*, bottom sketch.

## Non-linear dependency on myosin levels

In general, the active torque density $\tau$ and the active tension $T$ are functions of the myosin density, which in turn is proportional to NMY-2::GFP fluorescence intensity $I(x)$. In the absence of myosin, $T$ and $\tau$ vanish. For increasing $I$, we express active tension and active torque density in a Taylor expansion,

$$T = \alpha_0 I + \alpha_1 I^2 + O(I^3),$$

$$\tau = \beta_0 I + \beta_1 I^2 + O(I^3), \tag{20}$$

with coefficients $\alpha_0$, $\alpha_1$, $\beta_0$, and $\beta_1$. The chirality index $c = \tau/T$ then shows a myosin dependence of the form,

$$\tau/T \simeq \frac{\beta_0 + \beta_1 I}{\alpha_0 + \alpha_1 I} \simeq \frac{\beta_0}{\alpha_0}\left(1 + \left(\frac{\beta_1}{\beta_0} - \frac{\alpha_1}{\alpha_0}\right)I\right), \tag{21}$$

where we have neglected higher order terms. Rho signalling may directly influence the levels of myosin and thereby $I$. Also, Rho signaling affects the structure of the actin cytoskeleton (*Figure 3—figure supplement 4*), and could alter the fraction of active myosin, possibly even in a position-dependent manner. Rho exhibits a spatial profile similar to that of NMY-2 (*Motegi and Sugimoto, 2006*), which is captured by the function $I(x)$. The non-linearities induced by Rho can thus be accounted for by non-linearities in $I(x)$. We can therefore approximate such additional effects of Rho signaling by modifications of the position-independent parameters $\alpha_0$, $\alpha_1$, $\beta_0$, and $\beta_1$.

However, our experimental data do not allow us to faithfully estimate the values of $\alpha_1$ and $\beta_1$. Thus, we consider only linear dependences in our fitting procedure using $T = \alpha I$ and $\tau = \beta I$. The values for $\alpha$ and $\beta$ and thus the chirality index $c = \alpha/\beta$ that our fitting routine determines correspond to averages of $\alpha_0 + \alpha_1 I$ and $\beta_0 + \beta_1 I$ along the embryo.

## Fitting phenomenological coefficients

To compare our theory to experiment, we numerically solve *Equations 17* and *18* using a finite difference scheme, using the two extreme points of the measured velocity profile as boundary conditions. *Equations 17* and *18* depend on three free parameters, the hydrodynamic length $l$, and the constants $\alpha$ and $\beta$ which relate the measured fluorescence intensity $I$ to the active tension and the active torque density such that $T = \alpha I$ and $\tau = \beta I$, respectively. We adjust these parameters by performing a least squares fit of the solutions to *Equations 17* and *18* to the measured velocity profile. In this way we obtain values for the hydrodynamic length $l$ and the chirality index $c = \tau/T$ numerically. We obtain error estimates for $l$, $\alpha$, and $\beta$ from the Hessian of the residual of the least square fit with respect to the parameters.

## Acknowledgements

We are grateful to S Schonegg, A A Hyman, and W B Wood for their observation and analysis of the chiral whole-cell rotation of the dividing *C. elegans* zygote, without which we would not have recognized the importance of observing chiral cortical flow at an earlier stage. We are also grateful to B Fievet, J Rodriguez, and J Ahringer for observations of chiral flow in a PAR suppressor screen. We would also like to thank J S Bois, M Labouesse, J Prost, S B Reber, K Vijay Kumar, A-C Reymann, P Gross, P Chugh, and D Needleman for valuable discussions and critical comments. This work was supported by grant no. 281903 from the European Research Council (ERC) and SF acknowledges DFG and Human Frontier Science Program (HFSP) for funding.

## Additional information

### Competing interests

FJ: Reviewing editor, *eLife*. The other authors declare that no competing interests exist.

### Funding

| Funder | Grant reference number | Author |
| --- | --- | --- |
| European Research Council | 281903 | Sundar Ram Naganathan, Masatoshi Nishikawa |
| Deutsche Forschungsgemeinschaft | FU-961/1-1 | Sebastian Fürthauer |
| Human Frontier Science Program | Cross disciplinary fellowship LT000871/2014 | Sebastian Fürthauer |

The funders had no role in study design, data collection and interpretation, or the decision to submit the work for publication.

## Author contributions

SRN, Conception and design, Acquisition of data, Analysis and interpretation of data, Drafting or revising the article; SF, SWG, Conception and design, Analysis and interpretation of data, Drafting or revising the article; MN, Conception and design, Analysis and interpretation of data; FJ, Analysis and interpretation of data, Drafting or revising the article

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
