## [Decision Letter]

Thank you for sending your work entitled “Active torque generation by the actomyosin cell cortex drives chiral symmetry breaking” for consideration at *eLife*. Your article has been favorably evaluated by Janet Rossant (Senior editor), a Reviewing editor, and 2 reviewers.

These results are interesting and timely as they provide strong experimental and theoretical support to a role of the actin cytoskeleton in breaking L/R symmetry in early stage embryos. Although this idea has been supported by some data on a few model organisms (including a recent paper by Schonegg et al., Genesis 2014 describing the chiral flow at the 1-cell stage and showing its dependence on nmy-2 activity) and has been discussed in recent reviews, the findings and concepts reported here provide strong advances to the field with a focus on the quantitative characterization of the chiral flows. The results point towards a significant connection between actomyosin-based chiral flows and LR symmetry breaking at embryo/tissue scales, with implications for LR symmetry-breaking and chiral morphogenesis in many other contexts.

The Reviewing editor and the other reviewers discussed their comments before we reached this decision, and the Reviewing editor has assembled the following comments to help you prepare a revised submission. We have one major point and a number of minor points for you to consider when revising your manuscript.

Major point:

1) The authors are inferring a connection between chiral flows in the zygote, those that occur at the 4-cell stage in ABa and ABp, and the skew of ABa and ABp that represents the first observable manifestation of embryonic handedness. But the links between observations made in the zygote and those at the 4-cell stage, remain tenuous. What would greatly strengthen this connection in my opinion would be to document a correlation between effects on chiral flow and skew - both measured at the 4-cell stage, for a few key perturbations (e.g. weak ect-2(RNAi), weak rga-3./4(RNAi, gsk-3(RNA) and possibly one of (dsh-2, mig-5, mom-2). My sense is that this could be done with relatively little additional effort.

Minor points:

2) The quantitative analysis of chiral flow and its interpretation in terms of a physical model for active chiral fluids is nicely done, but we think the authors need to tone down the claim that they have “discovered that the actomyosin cytoskeleton generates active torques...” What they have in fact shown is that a physical model assuming active torques can reproduce the observed kinematics.

3) It would be nice if the authors could comment on two related points:

(a) In the physical theory, counter-rotating flow depends on a gradient of active torque density. In the zygote this gradient is presumed to arise from a gradient of Myosin II activity, accounting for the simultaneous existence of AP and counterrotating flow. What about at the 4-cell stage? Are there axial flows (and possibly gradients of Myosin density) that coincide with the reported counter-rotation? Is the overall pattern of chiral + axial flow consistent with the physical theory?

(b) A basic assumption of the physical model is that actomyosin interactions have an intrinsic handedness that determines the handedness of the active local torque, which in turn determines the handedness of counter-rotating flow. But what happens when LR asymmetry is reversed as in cold-treated embryos or certain mutants? Is there a change in the handedness of the flow? Or (a more likely scenario) does this imply that there are other processes that can completely override any contribution of chiral flow to dictate handedness. We are not asking that the authors resolve this issue experimentally, but it would be nice if they could discuss briefly the implications.

4) When referring to Drosophila work, the authors should cite the 2 original Nature papers characterizing the myosinID gene: [16] and Speder et al. 2006. In their reference list, the journal for Hozumi et al. is not spelled out correctly (should be Nature instead of Nat Cell Biol).

5) The chiral flows are intracellular by nature suggesting a cell autonomous effect. However the text is confusing at it is stated that the origin of the skew comes from 1-cell stage while concluding that it originates from chiral torque during 4-cell stage. The specific contribution of each flow (1-cell vs 4-cell) on the establishment and/or maintenance of handedness is thus unclear. Uncoupling the contribution of each flow would clarify the model and more clearly address the origin and duration of the effect of the flow on later events. Of note, results from Schonegg et al. suggest an independent origin of embryo rotation and handedness (see Table 2 in Schonegg et al.).

6) The authors mention that they could not find any defect in early embryos upon perturbing the flow at the 1-cell stage. We presume they have looked at embryo cortex rotation (and spindle asymmetry) and if so, their findings seem to contradict results from Schonegg et al. showing that defective acto-myosin network blocks early rotation of the 1-cell zygote (see Table 1 in Schonegg et al.). Comments?

7) Although they raise the question, the authors do not discuss what may control flow directionality, which is key to understanding symmetry breaking? Could it be asymmetry of the actin network, intrinsic molecular chirality of myosin and actin, etc? I think that expanding the discussion on this point would help clarify and complement the model.

8) Could the authors explain what in their view could be the basis of the independence of contractile tension (along A/P axis) and torque density (along L/R axis)?

9) Could the authors be more clear on how could a directional flow at the cortex induce asymmetry, for example of the spindle?

---

## [Author Response]

*1) The authors are inferring a connection between chiral flows in the zygote, those that occur at the 4-cell stage in ABa and ABp, and the skew of ABa and ABp that represents the first observable manifestation of embryonic handedness. But the links between observations made in the zygote and those at the 4-cell stage, remain tenuous. What would greatly strengthen this connection in my opinion would be to document a correlation between effects on chiral flow and skew - both measured at the 4-cell stage, for a few key perturbations (e.g. weak ect-2(RNAi), weak rga-3./4(RNAi, gsk-3(RNA) and possibly one of (dsh-2, mig-5, mom-2). My sense is that this could be done with relatively little additional effort*.

This is a very good suggestion. As requested, we have now quantified chiral flow velocities for wild type, *ect-2* and *rga-3* RNAi conditions in the ABa cell. We find that changes in skew angle are indeed concomitant with changes in chiral counterrotation velocity vc in ABa, with *ect-2* (vc: -3.4 ± 1.4 μm/min, n=8, skew: 15.8 ± 4.9°, n=14) RNAi leading to reduced chiral flows and a reduced skew, and *rga-3* (vc: -6.7 ± 0.7 μm/min, n=8, skew: 37.8 ± 6.1°, n=13) leading to increased chiral flows and an increased skew compared to wild type (vc: -5.2 ± 1.1 μm/min, n=10, skew: 23.6 ± 3.7°, n=12).

We have included a new plot, Figure 4, with wild type, *ect-2* and *rga-3* chiral velocity quantifications and have updated Video 6 that displays chiral flow under *ect-2* and *rga-3* RNAi conditions in addition to the non-RNAi condition. An in-depth analysis of all Wnt pathway members and their control of chiral flow, together with a mechanistic picture of how chiral flows execute the clockwise skew event warrants an independent analysis, which is beyond the scope of our manuscript.

However, the new Figure 4 allows us to conclude that active torque generation and chiral counterrotatory flows participate in execution of left-right (LR) symmetry breaking of the embryo.

*2) The quantitative analysis of chiral flow and its interpretation in terms of a physical model for active chiral fluids is nicely done, but we think the authors need to tone down the claim that they have “discovered that the actomyosin cytoskeleton generates active torques...” What they have in fact shown is that a physical model assuming active torques can reproduce the observed kinematics*.

We agree with this suggestion and have toned down the wording accordingly.

3) It would be nice if the authors could comment on two related points:

(a) In the physical theory, counter-rotating flow depends on a gradient of active torque density. In the zygote this gradient is presumed to arise from a gradient of Myosin II activity, accounting for the simultaneous existence of AP and counterrotating flow. What about at the 4-cell stage? Are there axial flows (and possibly gradients of Myosin density) that coincide with the reported counter-rotation? Is the overall pattern of chiral + axial flow consistent with the physical theory?

This is a very good point, we do observe axial flows at the 4-cell stage. The cleavage plane comprises a stripe of high myosin activity leading to a gradient on either side of the stripe (Video 6). This will lead to both chiral flows and axial flows as observed in our movies at the 4-cell stage. However, quantification of axial flows is non-trivial because of an additional rotational movement (Figure 4—figure supplement 1), which also enters our quantification as an axial flow (when viewed from the dorsal side of the embryo).

A full quantification of the different flows generated at the 4-cell stage is beyond the scope of the paper. Please note that this would require a precise analysis of cell-cell interactions with neighbors. However, on a qualitative level, the myosin distributions in ABa and ABp (Video 6) are such that the overall pattern of chiral and axial flow is consistent with the predictions from our physical theory.

The expected flow profiles from our theoretical description given a stripe of high myosin activity (corresponding to the cleavage plane) is shown in newly added Figure 4—figure supplement 5.

*(b) A basic assumption of the physical model is that actomyosin interactions have an intrinsic handedness that determines the handedness of the active local torque, which in turn determines the handedness of counter-rotating flow. But what happens when LR asymmetry is reversed as in cold-treated embryos or certain mutants? Is there a change in the handedness of the flow? Or (a more likely scenario) does this imply that there are other processes that can completely override any contribution of chiral flow to dictate handedness. We are not asking that the authors resolve this issue experimentally, but it would be nice if they could discuss briefly the implications*.

This is a very interesting question. In the case of cold treatment, a 0.5% increase of sinistral handed embryos was observed by Wood WB et al. (1996) when the worms were maintained at 10°C. It has to be noted that the myosin ATPase activity and actin filament sliding velocity also reduce by 4-fold at low temperatures (Yanagida T et al., Nature, 1996). Therefore, we would expect a marked reduction in myosin driven activities and significantly reduced chiral flows in cold-treated embryos. Thus, in the case of cold-treated embryos, because chiral flows are likely to be significantly reduced, additional unknown processes could effect a marginal increase in sinistral handed embryos.

Similarly, in the case of certain mutants such as *gpa-16(it43)* reported by [2], it is likely that the handedness of flow is unaffected.

Given that *gpa-16* genetically interacts with a few *par* genes (2), it is possible that cortical localization of myosin is impaired, which could affect chiral flow. Also, the skew event facilitated by chiral flows is likely to be dependent on neighboring cell-cell interactions as well (see response to point 9). Given the aberrant behavior of mitotic spindles in the first three cleavages prior to LR symmetry breaking (2), the *it143* mutant could have slightly misplaced neighbors that could as well result in impaired LR symmetry breaking.

It remains to be seen which processes act in concert with chiral flows to dictate embryonic handedness and as the reviewers have indicated we also expect chiral flow handedness to be unaffected in cold-treated embryos as well as the different mutants.

*4) When referring to Drosophila work, the authors should cite the 2 original Nature papers characterizing the myosinID gene:*
[16]
*and*
*Speder et al. 2006**. In their reference list, the journal for Hozumi et al. is not spelled out correctly (should be Nature instead of Nat Cell Biol)*.

We have included the suggested paper in the reference section and corrected the Hozumi et al., reference.

*5) The chiral flows are intracellular by nature suggesting a cell autonomous effect. However the text is confusing at it is stated that the origin of the skew comes from 1-cell stage while concluding that it originates from chiral torque during 4-cell stage. The specific contribution of each flow (1-cell vs 4-cell) on the establishment and/or maintenance of handedness is thus unclear. Uncoupling the contribution of each flow would clarify the model and more clearly address the origin and duration of the effect of the flow on later events. Of note, results from Schonegg et al. suggest an independent origin of embryo rotation and handedness (see Table 2 in Schonegg et al.)*.

We are sorry for the misunderstanding and we have amended the conclusions in the manuscript to read better. We did not intend to mean that the origin of the skew comes from the 1-cell stage, but rather wanted to state that there is a general link between LR symmetry breaking genes and chiral flows. We completely agree with Schonegg et al., in the fact that cortical chirality independently regulates early chiral events at the 1-cell stage as well as the chiral event at the 4-cell stage.

Chiral flow generation is an intrinsic property of the actomyosin cortex and is generated independently at multiple stages during *C. elegans* development.

*6) The authors mention that they could not find any defect in early embryos upon perturbing the flow at the 1-cell stage. We presume they have looked at embryo cortex rotation (and spindle asymmetry) and if so, their findings seem to contradict results from Schonegg et al. showing that defective acto-myosin network blocks early rotation of the 1-cell zygote (see Table 1 in Schonegg et al.)*. *Comments?*

We apologize for the poor wording in this part of the main text. We intended to comment that counterrotatory flows during polarity establishment at the 1-cell stage cannot provide information about LR symmetry breaking of the embryo because of the presence of just a single established body axis.

We have corrected the main text accordingly. It is important to understand that the chiral counterrotatory flows described in this paper observed before pronuclei meeting at the 1-cell stage and during the 4-cell stage, are mechanistically different from the chiral flows that lead to a whole body rotation during cytokinesis of the 1-cell stage described in Schonegg et al.

The latter event is also likely to require the generation of clockwise torques, however, the physical basis of this whole body rotation during cytokinesis is not understood.

*7) Although they raise the question, the authors do not discuss what may control flow directionality, which is key to understanding symmetry breaking? Could it be asymmetry of the actin network, intrinsic molecular chirality of myosin and actin, etc? I think that expanding the discussion on this point would help clarify and complement the model*.

At the 1-cell stage, when viewed from the posterior pole, we observe the flow direction to be counterclockwise in the posterior and clockwise in the anterior. This directionality is due to the generation of clockwise torques (when viewed from the outside) in the cortical layer.

A gradient in these torques then leads to either counterclockwise or clockwise flows depending on the direction of the gradient as illustrated in Figure 2, bottom sketch. It will now be interesting to investigate the molecular basis of clockwise torques and as the reviewers have indicated, the observed chirality is likely due to a combination of an intrinsic chirality of the actin filament, a mismatch in the step size of myosin and the helical pitch of actin and a symmetry broken orthogonal to the cortical plane with the membrane on one side and the cytoplasm on the other side. A clarification of this is beyond the scope of our current work.

8) Could the authors explain what in their view could be the basis of the independence of contractile tension (along A/P axis) and torque density (along L/R axis)?

This is a very good point that was not properly discussed. We have demonstrated that myosin molecular motors are the origin of both active tension and active torque in the cortex. There are two possibilities wherein (i) myosin either directly produces torque (dipoles) or (ii) myosin generated force (dipoles) are converted to torque dipoles via the structure of the actin network.

Our finding that active torques can be regulated independently from active tension hints at the latter possibility (ii). A definitive answer is however, beyond the scope of the present study.

9) Could the authors be more clear on how could a directional flow at the cortex induce asymmetry, for example of the spindle?

We propose that the mechanism by which chiral flows execute the skew is similar to a crawler excavator or a digger with both chains rotating in opposite directions, allowing the whole machine to turn on the spot.

Chiral counterrotatory flows in the ABa and ABp daughter cells with the help of adhesion to the substrate below (EMS cell) would then cause the daughter cells to roll past each other pushing the left daughters more anterior than the right. A corresponding and clarifying sentence has been added to the manuscript.